# The Application of Hyperspectral Imaging to the Measurement of Pressure Injury Area

**DOI:** 10.3390/ijerph20042851

**Published:** 2023-02-06

**Authors:** Lin-Lin Lee, Shu-Ling Chen

**Affiliations:** Department of Nursing, Hungkuang University, Taichung 433304, Taiwan

**Keywords:** hyperspectral imaging, pressure injury, area measurement

## Abstract

Wound size measurement is an important indicator of wound healing. Nurses measure wound size in terms of length × width in wound healing assessment, but it is easy to overestimate the extent of the wound due to irregularities around it. Using hyperspectral imaging (HIS) to measure the area of a pressure injury could provide more accurate data than manual measurement, ensure that the same tool is used for standardized assessment of wounds, and reduce the measurement time. This study was a pilot cross-sectional study, and a total of 30 patients with coccyx sacral pressure injuries were recruited to the rehabilitation ward after approval by the human subjects research committee. We used hyperspectral images to collect pressure injury images and machine learning (k-means) to automatically classify wound areas in combination with the length × width rule (LW rule) and image morphology algorithm for wound judgment and area calculation. The results calculated from the data were compared with the calculations made by the nursing staff using the length × width rule. The use of hyperspectral images, machine learning, the length × width rule (LW rule), and an image morphology algorithm to calculate the wound area yielded more accurate measurements than did nurses, effectively reduced the chance of human error, reduced the measurement time, and produced real-time data. HIS can be used by nursing staff to assess wounds with a standardized approach so as to ensure that proper wound care can be provided.

## 1. Introduction

Pressure injury (PI) development is a common wound care problem in chronically ill patients, the elderly, and patients with limited mobility in acute wards, nursing homes, long-term care, or home care [1]. The prevalence in long-term care facilities ranges from 2.3% to 28%, a clear indication of its importance [2]. PIs can lead to patient discomfort, decreased quality of life, prolonged hospital stay, and increased financial burden. In addition, they negatively impact quality of care and increase nursing workforce and healthcare costs, so healthcare professionals focus on PI prevention [3,4]. The problem of chronic wounds is also becoming more and more complex, making it more challenging for nurses to face wound problems clinically. Nurses, in particular, are the primary assessors and caregivers of patients with crush wounds. Wound care is no longer a matter of simply changing a dressing. The nature of the wound should be deeply understood, including the wound assessment, measurement, healing mechanism, provision of an effective wound healing environment, and prevention of complications [5]. In addition to the professional knowledge of wound care, nurses also need to have the professional ability to perform accurate wound assessments and describe wound changes as the basis for wound treatment [6]. In the clinical measurement of pressure wounds, plastic rulers in centimeters are commonly used to measure the surface of a wound by calculating its length × width. Since the wound is often irregular in shape, it is easy to overestimate the extent of the wound, which can affect the consistency of the measurement [7]. Studies have pointed out that the wound area obtained by the length × width rule is often overestimated by about 44%, resulting in serious inaccuracies and uncertainties in judging the degree of wound recovery of patients [8,9]. Therefore, accurate wound measurement is a necessary preliminary step to providing good care. In addition, the use of a standardized method to assess wounds would eliminate subjective bias and ensure accurate assessments of their severity. It would also provide accurate wound measurement information [6].

Despite the fact that the knowledge and judgment of clinical staff are essential for every type of clinical diagnosis, including pressure sores, such judgments are affected by subjective bias. To address this issue, some researchers have focused on automated wound assessment as a complement to traditional visual inspection. With the rapid development of science and technology, technology can assist in wound management. Yu et al. [5] pointed out that future wound management will develop towards intelligence and mobility. Wound caregivers such as nurses can use images of wounds to automatically interpret and complete wound assessment records such as wound size, infection status, and healing status.

Hyperspectral imaging (HSI) is a new technology for noninvasive, noncontact, and automated optical measurement [10]. HSI is a very common technology in the field of telemetry, and it has been applied in agriculture [11], mining [12], food quality [13,14], uniform analysis of pharmaceutical ingredients [15], environmental monitoring [16], biomedicine [17], and other fields. Hyperspectral imaging technology combines the original spatial information and spectral wavelength information. The obtained spatial data provide a two-dimensional image, and the intensity of the received light is converted into a photoelectric signal. The data are composed of three dimensions, two of which are the spatial dimensions of the length and width of the photo. The third dimension is composed of the image data of different wavelengths (Figure 1).

In recent years, the use of HSI in the medical field in Western countries has become a special topic of research. Research on HSI in skin diseases has focused on skin burn and scald healing evaluation [18], skin melanoma [19], the measurement of tissue blood oxygenation, and prediction of the risk of diabetic foot ulcer formation [10] and pressure injury [20]. Datasets have been combined with machine learning for purposes such as medical data diagnosis, face recognition, data classification, and trend prediction. Algorithms are used to classify large amounts of collected data or to train prediction models. Once a good model is built, new data can be used to make predictions with the trained model. The advantage of unsupervised learning is its usefulness in the early stage of data mining. In the early stage, the concepts of classification and grouping can be automatically developed according to the characteristics of the data without prior knowledge of the answer. Compared with supervised learning, unsupervised learning can greatly reduce the amount of tedious manual work.

Our study used data from HSI and machine learning to assess pressure injuries and compared the accuracy of HSI with length × width wound measurements obtained by nursing staff. This method of clinical wound assessment can complement traditional visual inspection and measurement with optical techniques [20].

## 2. Method

### 2.1. Research Design

This study was a pilot cross-sectional study with a prospective design. Thirty coccygeal PI patients were recruited from the rehabilitation ward of a teaching hospital. We collected wound images using a hyperspectral detector and performed wound area calculations using unsupervised learning and morphological algorithms.

After the wounds were located with the automated program, the data were pre-processed. In the first step, the program was used to frame the wound area and delete redundant data, defined as non-related wound information such as bed sheets and clothing. After the wound area was framed, the k-means algorithm (k-means) for unsupervised learning was used to automatically distinguish the hyperspectral data of healthy skin and wounds.

The main function of the k-means algorithm is to classify similar data into one category. Each data point can be classified into a group, and each group is called a cluster. As for the classification rules, the distances between data points are used for calculation. The characteristics of each data point are represented by vectors, and each point is classified with the cluster center closest to itself.

In clinical nursing, most nurses measure the area of crush wounds visually or with a measuring ruler to obtain the length and width of the wound, and then they multiply the length by the width to calculate the area, which is called the length × width rule (LW rule).

### 2.2. Measurements

To collect data on pressure-damaged tissue, HSI data were captured with an HSI device with a snapshot mosaic hyperspectral image sensor (IMEC Inc., Ghent, Belgium), which has 16 bands: 465, 474, 485, 496, 510, 522, 534, 546, 548, 562, 578, 586, 600, 608, 624, and 630 nm. The AI program was used to calculate the pressure injury area from the spectral imaging data, and the results were also compared with the pressure injury areas measured by nurses for reliability analysis with some statistical parameters. To obtain the HSI data, the researchers followed a standardized experimental protocol: (1) a ruler was used to measure the size of the wound area; (2) the HSI test was explained to the patient; (3) the HSI device was calibrated by focusing it on a whiteboard with a reverse color ratio of 98%, and this data served as the benchmark value for the imaging environment; (4) the imaging location on the patient was confirmed, images were captured at a distance of 70 cm, and HSI data were obtained; (5) the pressure wound area was calculated from the HSI image by the AI program.

### 2.3. Data Collection

The data collection period was from 5 December 2019 to 25 April 2020. The researchers captured hyperspectral images of PIs on the tail and sacrum of 30 patients in the rehabilitation ward. An ideal environment for hyperspectral imaging is one without interference from extraneous light sources. However, since the subjects to be photographed in this research had restricted mobility, the hyperspectral images needed to be captured in different environments. To overcome the environmental challenges as much as possible, hyperspectral white correction was performed on each hyperspectral image so that the hyperspectral image data would be consistent and the error of the subsequent analysis would be reduced. In the same session, a nurse used a plastic ruler to measure the length and width (cm).

### 2.4. Statistics and Data Analysis

Basic characteristics were analyzed in SPSS Statistics version 22.0 (SPSS Inc., Chicago, IL, USA). SigmaPlot 12.5 and SAS 9.4 were used for statistical analysis, including the single statistic method for groups and repeated statistical measurements for individual patients. Meanwhile, statistical parameters such as Pearson’s correlation coefficient (*r*)*,* Spearman’s ranked correlation coefficient (*ρ*), the intraclass correlation coefficient (ICC), and unweighted and weighted kappa values (κ) were used to compare performances between the pressure injury areas determined from hyperspectral images and the pressure injury areas measured by the nursing staff. The clinical classification of pressure sore in this study was based on the standard classification of the National Pressure Ulcer Advisory Panel (NPUAP), which includes the gradations of Stage 1, Stage 2, Stage 3, Stage 4, and Unstageable. All patient cases were included in the analysis and all data were analyzed in triplicate.

### 2.5. Ethical Considerations

Approval was obtained from the Regional Teaching Hospital Human Research Committee (Review No. -108-75) prior to commencing the study. Eligible patients or their guardians were provided with a statement of the purpose and methods of the study, and all who agreed to participate in the study signed an informed consent form. To protect their privacy, the data collected were archived with numerical identifiers. If patients or their guardians wished to terminate their participation in the study, they could withdraw at any time and the data would be deleted immediately.

## 3. Results

### 3.1. Demographic Data of Patients

Table 1 shows that the mean age of the participants was 71.7 ± 15.97 years, and 20 (66.7%) of the patients were male. The most common admission diagnosis was pneumonia (26.7%), followed by cerebrovascular disease (16.7%), chronic respiratory failure (13.3%), and diabetes (10.0%). The crushed parts were all in the tail sacrum. There were 2 cases (6.67%) of the first stage, 13 cases (43.33%) of the second stage, 10 cases (33.33%) of the third stage, and 1 case (3.33%) of the fourth stage. Four cases (13.33%) could not be staged. Table 2 presents the classifications of pressure injury of patients judged by the doctors in this study.

Table 3 lists the results of the length × width rule for nurses, the length × width rule for machine learning, and the image morphology for calculating wound area. From the actual hyperspectral images and patient data, it was found that, when the wound area was small (such as Case 01, Case 04, Case 14, Case 29, and Case 30), the wound sizes measured by nursing staff were similar to those measured by machine learning. In cases with large wound areas, such as Case 08, Case 10, Case 19, and Case 22, the measurements of the nursing staff were different from those of the machine learning method. In addition, the actual wound characteristics were effectively distinguished by the machine learning method, revealing that the nursing personnel integrated the entire red and swollen area and the actual wound into the “wound area”. The cases marked with asterisks indicate that the wound measurements were determined by both machine learning and nurses. Due to the inconsistency of the measurement positions, the calculated results were significantly different.

### 3.2. Hyperspectral Image Data

HSI data from 30 patients were included in the analysis. Non-invasive HSI was used to capture images of patients with coccyx sacral crush injuries (Figure 2), and the information obtained was used for the unsupervised learning of the machine learning algorithm (k-means) to analyze the wound area. In Figure 1, the red area is the crush wound area, and each grid of the red dotted line is 1 cm^2^. The area of the red crush wound, calculated by using the LW rule and the image morphology algorithm, was 7.9 cm^2^ (Figure 3).

### 3.3. Comparison of Related Statistical Parameters of Various Wound Area Methods

The validity was tested with Pearson’s correlation coefficient (*r*) and Spearman’s correlation coefficient (*ρ*), and the intragroup correlation coefficient (ICC) and weighted kappa value (κ) were used as the reliability indicator and the consistency indicator for the single statistic method, respectively.

The statistical results indicated a significant correlation and consistency between the length × width rule (LW) results of the nursing staff and the length × width rule (LW) results of the machine learning method (*r* = 0.82, *p* < 0.001; *ρ* = 0.82, *p* < 0.001; ICC = 0.81, *p* < 0.001; κ = 0.76, 95% CI 0.5–0.97), but after further comparison with machine learning combined with morphological results, the correlation coefficient dropped significantly (*r* = 0.44, *p* < 0.05; *ρ* = 0.44, *p* < 0.05; ICC = 0.54, *p* < 0.05, κ = 0.42, 95% CI 0.06–0.78). However, after observing the actual wounds, we found that, in some cases, the change could be related to inconsistencies in the determination of the location of the wound. The deviation above might have been related to the self-judgment logic error of the AI algorithm due to interference from the light source. At this time, the judgment of wound location should still be performed by trained medical personnel. The data in Table 4 also show that if the seven cases (Cases 02, 06, 07, 21, 23, 24, and 28) with inconsistent determination of wound location were considered outliers and omitted, the results of the length × width rule (LW) calculation of the nursing staff and the machine were better. The correlation coefficients of the operation results of the length × width rule (LW) of the machine learning style all improved and had statistical significance (*p* < 0.001). In the machine learning method, the length × width rule (LW) was more relevant than the morphological judgment rule, which also showed that the size of the wound area could indeed be measured with noninvasive hyperspectral images, which could effectively reduce nurses’ workloads or errors.

In addition, statistical analysis methods for repeated measurements were performed in SAS software, and linear mixed models were also applied (The variable Subject was set as the class item and random effect). As shown in Table 5, the ICC outcomes of the combination of machining learning and morphology were still better than those of the methods using machine learning technology alone (ICC outcomes: 0.8372 and 0.7728, respectively) in the presence of outliers. The utility of measuring the wound area using the HSI device with the combination of machine learning and morphology was evident.

### 3.4. Difference in Measurement Time for Pressure Injury Assessment

The researchers used a snapshot hyperspectrometer to capture images of the PI area of the coccyx and the sacrum. It only took one second to capture a single image. The data were transmitted to a back-end computer within 90 s, and then the calculations were performed for 2 s to generate the data of the wound area. At the same time, a measurement ruler was used. For measuring the length × width (cm) area, the average time for PIs in phase 2 was 230–290 s, and the average time for PIs in phases 3 and 4 was 230–320 s. The time required for hyperspectral imaging of PIs was far less than that required for length × width measurement. In addition, the hyperspectral images could be transmitted electronically, without any physical writing. The most important effect is the reduction in the time required for wound area measurement, and simplifying the wound care evaluation process may reduce expenditures on medical supplies.

## 4. Discussion

The methods of measuring the area of pressure injuries used by nursing staff may vary depending on the facility or organization. However, determining the size of the wound typically involves visual inspection of the wound and the use of a measuring tool, such as a ruler or a wound measuring device. The wound measurement is then recorded in the patient’s medical chart. It is important to note that the measuring method used should be consistent and accurate to ensure proper tracking of the healing progress of the wound.

In this study, the nursing staff’s assessments differed from those of the machine learning method. After a detailed comparison of the imaging information, it was found that the machine learning method was able to effectively distinguish the actual characteristics of the wound, while nursing staff often categorized the entire red and swollen area and the actual wound as the “wound area”.

The results of the study showed that the noninvasive HSI technique can be used for the measurement of PI wound areas. In each case, the PI wound area was measured by the length × width rule (LW), and the PI wound was photographed by hyperspectral imaging. The area results were calculated by the length × width rule (LW) in the machine learning method, and the correlation and consistency between the two were compared. All had significant relationships (Pearson’s correlation coefficient = 0.82, *p* < 0.001; Spearman’s correlation coefficient = 0.82, *p* < 0.001; ICC = 0.81, *p* < 0.001, weighted kappa value = 0.76, 95% CI 0.55 to 0.97). However, compared with the judgment results of machine learning combined with morphology, the correlation coefficient and the degree of reproducibility dropped significantly (Pearson’s *r* = 0.44, *p* < 0.05; Spearman’s *ρ* = 0.44, *p* < 0.05; ICC = 0.54, *p* < 0.05; k = 0.42, 95% CI 0.06–0.78). Echoing the findings of Chiang et al. [8] and Langemo et al. [9], the actual area of the wound is different from the subjective, experience-based judgment of a nurse, and the wound area obtained by the length × width rule is often much larger than the actual area. Wound size may be overestimated by about 44%, and the extent of the wound is also easy to overestimate. In addition, echoing the research results of Luo et al. [21], wounds often appear irregular, and the wound depth is not calculated. The length × width (LW) rule is prone to overestimation of the extent of the wound, and wound photography can be used to estimate the wound area.

In addition, according to the systematic reviews of Saiko et al. [22] and Zahia et al. [23], objective wound size assessment by hyperspectral imaging has a high reference value for nursing staff, which was the reason why this study used hyperspectral imaging. The image data were used as the standard data of the wound size in this research, and then the wound area was calculated by morphology and compared with measurements made by the nursing staff using the two-dimensional length × width rule (LW). It was found that the visual method of the nursing staff was indeed insufficiently accurate.

Another large difference was found in measurement time. Hyperspectral imaging uses photographic imaging to judge the size of the wound, and the image analysis of each photo can be completed in less than one second. This aspect of hyperspectral imaging is also described by Saiko et al. [22], who mentioned that using the device, which is easy to operate and capable of rapid imaging, indeed required less measurement time than visual measurement by nursing staff (which takes at least 5 min). On the whole, the data of hyperspectral images can be transmitted without writing, just as physiological monitoring values and blood sugar levels can be transmitted. This digital process eliminates the need to write nursing records, and the data can be directly accessed with information technology.

The results of this study support the value of hyperspectral PI imaging and show that wound area calculated by morphology is more accurate than that measured by nurses using the two-dimensional length × width (LW) rule. The use of hyperspectral imaging can effectively reduce human error, achieve consistency in wound measurement, and reduce wound measurement time. Yu et al. [5] pointed out that wound assessment will develop towards intelligence and mobility in the future. Artificial intelligence can assist nurses by performing automatic wound image interpretation and complete wound size measurement, which will improve the accuracy of wound assessment.

## 5. Limitations

This study had several limitations. The first was that the study was conducted in a regional hospital, with a small sample size. Due to the bedridden status or limited mobility of most patients and the impact of the COVID-19 pandemic, only 30 PI patients could be examined in the six-month period of the study. The second limitation was that this study compared the measurements of the two-dimensional length × width rule (LW) commonly used in clinical practice with the length × width (cm) area calculated by hyperspectral image morphology. Finally, due to the limited data sources, this study could not provide data on wound areas measured in three dimensions (length × width × depth). The current research was a pilot study. It is hoped that based on the empirical basis of this research, further research can be carried out in medical centers with a larger number of samples.

## 6. Conclusions

Consistency in wound measurement is crucial in wound management. This study demonstrates that the noninvasive HSI technique can be used to measure wound size more accurately and in less time than traditional methods can, and also that it can provide consistency in wound assessment. This study also confirms that the digital imaging approach can automatically collect accurate wound information and records can be easily transmitted to electronic databases, simplifying the process for nurses and reducing measurement errors. The use of HSI is also an innovative development in the combination of nursing and technology, and it can be used both to obtain accurate information on wound healing progress and to provide medical staff with a basis for pressure wound treatment plans.

Future researchers may explore the following areas: (1) the use of hyperspectral imaging to actually calculate the area of irregular shapes so as to track changes in pressure wounds more effectively; (2) the use of hyperspectral imaging technology to conduct three-dimensional (length × width × depth) measurement of the wound area to collect more objective data; (3) the provision of hyperspectral imaging technology to clinical nurses for measurement of pressure injuries or other traumatic wound areas, with continuous evaluation of its effectiveness; and (4) the continued development of hyperspectral imaging technology for the assessment of pressure injury staging with cross-disciplinary teamwork.

## Figures and Tables

**Figure 1 ijerph-20-02851-f001:**
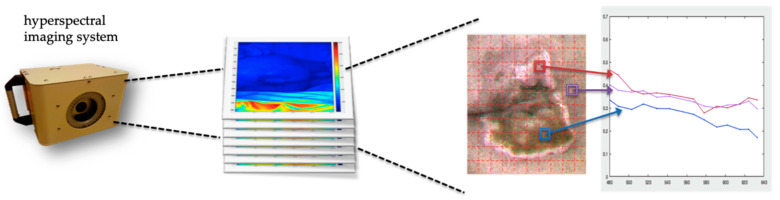
Hyperspectral Imaging (HSI) technology.

**Figure 2 ijerph-20-02851-f002:**
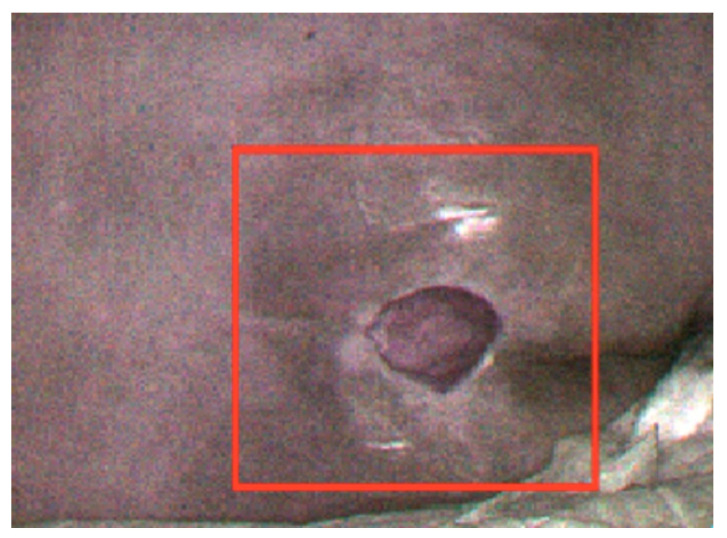
Hyperspectral image of coccyx sacrum pressure injury.

**Figure 3 ijerph-20-02851-f003:**
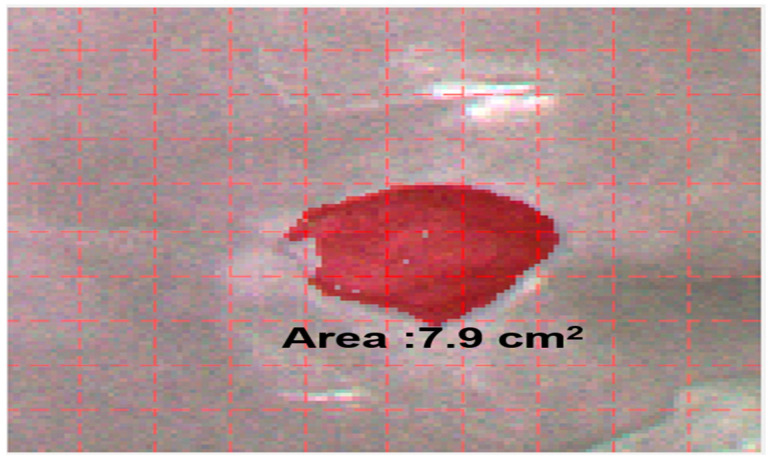
Coccyx sacral pressure injury with an area of 7.9 cm^2^.

**Table 1 ijerph-20-02851-t001:** Demographic characteristics of the cases (N = 30).

Variable	n	%
Gender		
Male	20	66.7
Female	10	33.3
Age (Mean ± SD), Y	71.7 ± 15.97	
Admission diagnosis Pneumonia	8	26.7
Respiratory failure	4	13.3
Obstructive lung disease	1	3.3
Cerebrovascular disease	5	16.7
Brain tumor	1	3.3
Intracranial injury	3	10.0
Diabetes Urinary tract infection Septicemia Buttock abscess Fever	3 3 2 1 1	10.0 10.0 6.7 3.3 3.3
Pressure injury stage		
Stage 1	2	6.6
Stage 2	13	43.3
Stage 3	10	33.3
Stage 4	1	3.3
Unstageable	4	13.3

**Table 2 ijerph-20-02851-t002:** The classifications of pressure injuries of cases judged by doctors (N = 30).

Subject	Classification
Case 01	Stage 2
Case 02	Stage 3
Case 03	Unstageable
Case 04	Stage 2
Case 05	Stage 3
Case 06	Stage 3
Case 07	Unstageable
Case 08	Unstageable
Case 09	Stage 4
Case10	Stage 3
Case 11	Stage 2
Case 12	Stage 2
Case 13	Stage 2
Case 14	Stage 3
Case 15	Unstageable
Case 16	Stage 2
Case 17	Stage 3
Case 18	Stage 3
Case 19	Stage 2
Case 20	Stage 2
Case 21	Stage 1
Case 22	Stage 3
Case 23	Stage 1
Case 24	Stage 2
Case 25	Stage 3
Case 26	Stage 2
Case 27	Stage 3
Case 28	Stage 2
Case 29	Stage 2
Case 30	Stage 2

**Table 3 ijerph-20-02851-t003:** The traditional length × width rule, machine learning with the length × width rule, and wound area information calculated by the image morphology algorithm.

Subject	Nurses Using the Length × Width Rule to Determine the Wound Area and Cal-culate the Area $\left(cm^{2}\right)$	The Machine Learning Meth-od Using the Length × Width Rule to Determine the Wound Area and Calculate the Area $\left(cm^{2}\right)$	Machine Learning Method Using Image Morphology Algorithm to Determine Wound Area and Calculate Area $\left(cm^{2}\right)$
Case 01	1.46	2.19	3.60
Case 02	12.05	10.22	22.60 **
Case 03	9.86	7.42	5.00
Case 04	0.18	0.09	0.30
Case 05	34.49	26.28	26.10
Case 06	4.93	5.11	20.70 **
Case 07	6.57	14.86	26.50 **
Case 08	54.75	40.88	26.80
Case 09	14.60	10.07	7.90
Case10	35.95	11.68	11.60
Case 11	23.66	11.41	7.00
Case 12	18.25	8.76	7.10
Case 13	10.95	8.41	8.80
Case 14	3.65	3.10	2.70
Case 15	39.79	36.73	34.40
Case 16	1.21	5.26	3.80
Case 17	39.89	35.77	35.40
Case 18	25.55	15.42	13.70
Case 19	59.13	35.04	25.70
Case 20	77.38	29.20	29.60
Case 21	1.64	7.01	39.10 **
Case 22	65.70	33.73	32.60
Case 23	0.04	0.73	27.20 **
Case 24	4.38	4.38	20.40 **
Case 25	20.44	13.14	12.10
Case 26	14.60	6.48	6.50
Case 27	24.09	21.90	16.10
Case 28	52.56	2.92	1.10 **
Case 29	3.07	0.61	0.50
Case 30	3.29	0.73	0.70

Note: ** = cases classified as outliers.

**Table 4 ijerph-20-02851-t004:** Comparison of related statistical parameters of various wound area methods.

	Wound Assessment Method	Staff Length × Width vs. Machine Learning Length × Width	Staff Length × Width vs. Machine Learning Combined with Morphology	Machine Learning Length × Width vs. Machine Learning Combined with Morphology
Statistical Parameters				
With Outliers				
Pearson’s correlation coefficient		0.80 *p* < 0.001	0.44 *p* < 0.05	0.69 *p* < 0.001
Spearman’s correlation coefficient		0.82 *p* < 0.001	0.44 *p* < 0.05	0.70 *p* < 0.001
Intraclass correlation coefficient ICC		0.81 *p* < 0.001	0.54 *p* < 0.05	0.81 *p* < 0.001
Unweighted kappa value κ		0.03 95% CI: [−0.02, 0.09]	0.01 95% CI: [0.01, 0.01]	0.01 95% CI: [0.01, 0.01]
Weighted kappa value Weighted κ		0.76 95% CI: [0.55, 0.97]	0.42 95% CI: [0.06, 0.78]	0.71 95% CI: [0.46, 0.95]
Without Outliers (The number of cases with inconsistent determination of wound location has been deducted)				
Pearson’s correlation coefficient		0.88 *p* < 0.001	0.87 *p* < 0.001	0.96 *p* < 0.001
Spearman’s correlation coefficient		0.93 *p* < 0.001	0.92 *p* < 0.001	0.97 *p* < 0.001
Intraclass correlation coefficient ICC		0.87 *p* < 0.001	0.85 *p* < 0.001	0.98 *p* < 0.001
Unweighted kappa value κ		0.01 95% CI: [0.01, 0.01]	0.01 95% CI: [0.01, 0.01]	0.01 95% CI: [0.01, 0.01]
Weighted kappa value Weighted κ		0.85 95% CI: [0.77, 0.94]	0.81 95% CI: [0.71, 0.91]	0.9560 95% CI: [0.92, 0.99]

Note: ICC = intraclass correlation coefficient; CI = confidence interval.

**Table 5 ijerph-20-02851-t005:** The ICC results of the linear mixed model for repeated measurements between two types of AI algorithm in the presence of outliers.

	Wound Assessment Method	Machine Learning Length × Width	Machine Learning Combined with Morphology
Statistical Parameters			
Intraclass correlation coefficient ICC		0.7728 *p* < 0.05	0.8372 *p* < 0.05

## Data Availability

The data are not publicly available due to restrictions regarding privacy and ethical considerations of the study participants.

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
