# Peer review of "The Application of Hyperspectral Imaging to the Measurement of Pressure Injury Area"

_ijerph, 2023, doi:10.3390/ijerph20042851_

Round 1
Reviewer 1 Report
The manuscript is well written and the topic is interesting but some points can be improved:
- the authors must specify the main features of the patients in particularly the comorbidites (diabetes mellitus and other comorbidities that can influence the wounds healing)
- the discussion is too short and the conclusion is too long: the authors must short the conclusion and focalize the main points while in the discussion the authors must include the limitations of the study and must discuss with recent literature the results
Reviewer 2 Report
The manuscript deals with the problem of PI (pressure sore) measurement in a clinical setting, and its reliability with the widely used measurement systems. The study recruited a little sample of inpatients of the rehabilitation ward, and measurements were compared within three methods.
Title: concise and clear, even if due to the (real) small sample size, “pilot cross-sectional study” specification need to be declared.
Introduction: main limitation of this section concern with the assumption that readers know what is Hyperspectral Imaging (HSI). Probably a more detailed description of the system could be helpful; for example, how the HSI-measurement are acquired, and processed. Only few (and obscure) specifications are presented in the methods, some images could help (at least in supplementary materials).
Lines 50-52, the meaning of this period is not clear, at least to me, what Authors would mean: “a standardized method to assess wounds would reduce cognitive differences”, in details I could not understand reduce cognitive differences in communication, and what does it means (also stated something similar in line 291).
Lines 54-57, personally I don’t agree with the meaning of this sentence, the assessment of pressure sore, as other clinical diagnosis, could not be only related to automatic evaluation, at least at the moment, a clinical (medical or nurse) judgement is needed. Could You discuss a little your position. As matter of fact, you stated in the objective declaration: “This method of clinical wound assessment can complement traditional visual inspection and measurement with optical techniques”.
Lines 74: what is a “train prediction models”? (Probably, this is my fault).
Method
Lines 89: clarify the study design, is a cross-sectional prospective study?
Lines 93-95: Does it means that you have a teaching-period for the machine or for the procedures, explain a little more.
Lines 111-116: You used several statistical approaches, but you used them to calculate the area from HSI or to compare the results from HSI to classic approach
Lines 115: do you have a standardized protocol to obtain HSI. For example, to take pictures of PS, during clinical trial were prepared support for camera, do you have something similar? Describe the standardization of measurement.
Lines 129: why do you use so many tests, please explain (in the text), otherwise the readers could image that you have tried several approaches.
In this section (method) miss also the classification of PS, and consequently we could also understand why and who are the four cases that are classified as unstageable, moreover, doe it means that those four subjects were not included in the analysis? Therefore, the sample dropped to 26 cases. Alternatively, if you use them in the analysis, probably those subjects could influence your results, please discuss in the limitation section.
Results
Table 2, is not so clear the text in the first row, can you describe the column title in footnotes.
Moreover, what ** means, is this a typo?
Lines 219-223: However, after comparing the images of actual patients (what does it means), we found that the change could be related to inconsistencies in the determination of the location of the wound in some cases. (Explain better, I have the sensation that in this sentence you are belittled something). What is inconsistent is not clear.
Table 3, is incomprehensible. You reported several statistics, but is not clear to me why. Could you try to use Linear mixed models to understand something more. You reported several comparisons between groups, without adjustment.
Lines 240-244 This sentence is more appropriate in the discussion.
Limitation
There are several limitations you have to discuss, for example:
Why a so short case finding
Do you have done a sample size calculation (why?)
3 Do you have a standardized acquisition protocol? Do you validate it with an inter-intrarater assessment
Etc…
Conclusion
Very speculative
Round 2
Reviewer 2 Report
The manuscript was significantly improved.
Good work.